# Ethanol Phase Change Actuator Based on Thermally Conductive Material for Fast Cycle Actuation

**DOI:** 10.3390/polym13234095

**Published:** 2021-11-24

**Authors:** Zirui Liu, Bo Sun, Jianjun Hu, Yunpeng Zhang, Zhaohua Lin, Yunhong Liang

**Affiliations:** 1State Key Laboratory of Mechanical Transmissions, Chongqing University, Chongqing 400044, China; 20162252@cqu.edu.cn; 2School of Mechanical and Aerospace Engineering, Jilin University, Changchun 130022, China; sunb19@mails.jlu.edu.cn; 3Key Laboratory of Bionic Engineering, Ministry of Education, Jilin University, Changchun 130022, China; zhangyunpeng17@mails.jlu.edu.cn (Y.Z.); liangyunhong@jlu.edu.cn (Y.L.)

**Keywords:** phase change materials, thermally conductive media, soft actuator, fast cycle actuation

## Abstract

Artificial muscle actuator has been devoted to replicate the function of biological muscles, playing an important part of an emerging field at inter-section of bionic, mechanical, and material disciplines. Most of these artificial muscles possess their own unique functionality and irreplaceability, but also have some disadvantages and shortcomings. Among those, phase change type artificial muscles gain particular attentions, owing to the merits of easy processing, convenient controlling, non-toxic and fast-response. Herein, we prepared a silicon/ethanol/(graphene oxide/gold nanoparticles) composite elastic actuator for soft actuation. The functional properties are discussed in terms of microstructure, mechanical properties, thermal imaging and mechanical actuation characteristics, respectively. The added graphene oxide and Au nanoparticles can effectively accelerate the heating rate of material and improve its mechanical properties, thus increasing the vaporization rate of ethanol, which helps to accelerate the deformation rate and enhance the actuation capability. As part of the study, we also tested the performance of composite elastomers containing different concentrations of graphene oxide to identify GO-15 (15 mg of graphene oxide per 7.2 mL of material) flexible actuators as the best composition with a driving force up to 1.68 N.

## 1. Introduction

As one of the miracles of nature, muscle, the source of human strength, has intensively inspired researchers to conduct a lot of research on soft deformation materials that develop and simulate the function of artificial muscle tissue [1,2]. Commonly used artificial muscles are mechanically driven, including pneumatic and hydraulic [3,4,5,6]. Various materials have been adopted to prepare the artificial muscles, which involves electroactive polymers [7,8,9,10,11,12], shape memory materials [13,14,15,16], hydrogels [17,18,19,20], and polymers driven by photothermal or humidity [21,22,23,24,25,26]. There are also biogenic actuators that rely on animal muscle cells [27,28]. As a drive for soft robots, fluid actuators require large external devices (e.g., air pumps, hydraulic pumps, etc.) to provide power, largely limiting the miniaturization of soft robotics applications. In contrast, electroactive polymers require high voltage (greater than a few kilovolts). Shape-memory alloys, shape-memory polymers and photo-thermal temperature-driven polymers require complex external stimuli (laser, humidity, and temperature control), and these actuators are difficult to adapt to more complex applications. Based on these, we believe that the ideal actuators need to have more direct and convenient actuation (current driven), safe actuation conditions (non-toxic and harmless to the operator), and fast and large strain capacity.

Currently, soft actuators based on phase change materials have received comparable interests, which mainly rely on mechanical forces generated by rapid expansion at phase change temperatures. Silicone rubber materials containing paraffin additives are an example of the integration of stimuli-responsive substances in an elastic matrix. Lipton and colleagues reported a volumetric expansion of approximately 10% for such composites using paraffin solid-liquid phase transitions [29]. Significantly higher swelling strains can be achieved by a reversible liquid-gas phase transition, but such material systems were difficult to control. Some recent actuators use entrained liquid vesicles or films to form an expanded cavity for actuation [30,31]. Electrically triggered deformation of soft elastic films using liquid-gas conversion of liquids has been reported to exhibit large area expansion [32]. Aslan and colleagues reported ethanol-gas phase transition composites with 900% expansion and correspondingly high stress (up to 1.3 MPa) and low density (0.84 g/cm^3^). The working mechanism, regenerative methods and the relationship between composition, structure and properties are thus, further characterized [33,34,35]. These phase change materials are excellent alternatives to conventional soft actuators, but the existing phase change materials still have some problems to be solved, such as low deformation rate, and difficult to control deformation, which are vital for the potential broad applications of the phase change materials.

In this work, we evaluated the functional properties of silicone/ethanol/(Au nanoparticle, graphene oxide) elastomer composites as a new approach to improvement and as a soft actuator. Ecoflex silicone elastomer is a kind of silicone rubber material commonly used for making soft robots, which has good flexibility and mechanical properties, and the model used herein is Ecoflex00-50. As a very common organic compound in daily life, ethanol is a flammable, volatile, colorless and a transparent liquid at room temperature and pressure with a boiling point of 78 °C. It is easy to realize vaporization after giving external heat and liquefying it to its initial state when the temperature decreases. The main source of deformation of the ethanol phase change drive is a gas/liquid change of ethanol, which is responsible for the volume change of the elastomer composites. Owing to the good electrothermal conversion rate, graphene oxide and Au nanoparticles were adopted as thermally conductive enhanced ingredients, respectively. The microstructure, mechanical properties, thermal imaging and mechanical drive characteristics of materials, containing no addition, addition of graphene oxide, and addition of Au nanoparticles, were characterized. Moreover, the influence of the addition of thermally conductive ingredients on drive efficiency was investigated. As part of the study, the mechanical properties and mechanical drive characteristics of composite elastomeric materials with different concentrations of graphene oxide were evaluated, to determine the optimized composition of the flexible actuator.

## 2. Materials and Methods

### 2.1. Material Synthesis

In this section, Ecoflex00-50 (Smooth-On Inc., Macungie, PA, USA) was used as the bulk material and ethanol (≥99.5%) was selected as the active phase change material. Both GO and Au nanoparticles were applied as the thermally conductive enhancement phase, respectively, because of their good electrothermal conversion performances. As shown in Figure 1a, for a control, ethanol was first added to the Ecoflex A component, then added to the B component to get the homogeneous solution (20 vol% of ethanol with respect to the 80 vol% Ecoflex mixture). The composite elastomer containing graphene oxide was also prepared via a similar experimental process, except for the adding of graphene oxide in the prepolymer solution. Au nanoparticle composite elastomers were prepared slightly different (firstly, aqueous solution of gold nanoparticles was thoroughly dried in a beaker, then dissolved in ethanol, placed in ultrasonic dispersion for 30 s, added to component A for about 2 min, and then mixed with component B for about 2 min). After the resistor wire was introduced, each of the mixture was thermally cured at room temperature for 5 h in a syringe mold to get the control ethanol phase change material. The specific formulation is shown in Table 1, where the concentration of Au nanoparticles is 1 mol/L. The material is poured into commercially available 5 mL syringes. After being fully cured, the composite elastomers were removed, and the properties of composites were tested (Figure 1b). Among those, a Ni-Cr resistive wire (d = 0.25 mm) was added to the composite cast mold for electrically driven heating of the artificial muscle. To facilitate the expansion of actuator material, the resistance wire was manually wrapped around plastic rod(d = 6 mm). Then we prepared composite elastomer with different heat conducting media, as shown in Table 2.

### 2.2. Driving Mechanism

Ecoflex00-50 is a silicone rubber material commonly used in the manufacture of soft robots, because of its good elasticity and mechanical properties. Ethanol is an organic volatile liquid with a boiling point of 78 °C. It vaporizes easily when external heat is applied and liquefies to its initial state when the temperature is lowered.

Ethanol, in the ethanol phase change artificial muscle exciter, is stored and dispersed in the silicone elastomer as vacuoles before start. When the voltage is turned on, the resistive wire in the ethanol phase change actuator is heated. As the temperature increases, ethanol bubble begins to gradually undergo a cavitation phase change from vacuole to bubble and the volume gradually expands, as shown in Figure 2. Due to the presence of nylon mesh, the McKibben-type ethanol phase change artificial muscle limits the axial expansion of the actuator. It can only expand radially and contract axially. After stopping its heating, ethanol bubbles return to their original vacuole. Due to the elasticity of silicone elastomer, the ethanol phase change artificial muscle returns to initial state, forming a telescoping reciprocal cycle of bi-directional actuation.

### 2.3. Micro-Topography Test

Scanning electron microscopy (SEM) is used for imaging and research of microstructures. Samples for SEM analysis were prepared by placing a composite sample (diameter 10 mm length 80 mm) in a container with liquid nitrogen. After 2 min, the sample was decomposed into 10 mm long irregular shapes, and the fragments were thawed.

### 2.4. Mechanical Properties

The composite material was made into a tensile specimen, with a length of 75 mm, a width of 10 mm and a height of 2 mm. A tensile test was carried out using a universal testing machine (C43, MTS Industrial Systems Co., Ltd., Eden Prairie, MN, USA) at a loading rate of 100 mm/min. Three sets of parallel tests were obtained to calculate the corresponding average value. The hardness test was carried out on the material using a hardness tester (model), and the average value was also obtained by an independent three sets of parallel tests.

### 2.5. Thermal Performance Test

Three composite materials (75 mm × 10 mm × 2 mm) were heated and energized simultaneously. The power supply was 15 v*3A. During the driving process, the thermal changes of the materials were recorded with a thermal imager (FLIR E4, FLIR Inc., Portland, OR, USA) every 10 s for a total recording time of 100 s. The images were processed, and the temperature analysis was performed by FLIR tools to determine the changes in material temperature.

## 3. Results and Discussion

### 3.1. Material Microstructure

We observed the microscopic morphology and the structure of the prepared ethanol phase change material by scanning electron microscopy. The experimental results are shown in Figure 3. It can be observed that the ethanol phase change actuator has a well-defined crater structure inside, which is produced by the vacuole formed by ethanol in silicone elastomer. The distribution of crater in ethanol phase change actuator is relatively uniform, indicating that the even distribution of ethanol in the ethanol phase change actuator. This uniform distribution is the basis for stable growth of its output force. It can be found that the interior of ethanol phase change actuator is relatively flat except for the pits. The shape of pits is hemispherical, which proves that ethanol exists in the form of spherical vacuoles in the ethanol phase change artificial muscle. The overall surface is relatively smooth with small and uniform wrinkles. We analyzed that the wrinkling was caused by the shrinkage of ethanol and silicone elastomer when they were frozen and cracked. The wrinkles inside pits are more intensive than those on the outer surface due to their spherical shape. The shrinkage of the inner surface caused by the rapid freezing and cracking of pits under liquid nitrogen is more pronounced than the shrinkage of the outer surface.

### 3.2. Deformation Property Test

The ethanol in the ethanol phase change actuator is stored and dispersed as a liquid bubble in the silicone elastomer. When the power is turned on, the resistive wire in the ethanol phase change actuator starts heating. As the temperature rises, the ethanol bubble begins to gradually change from a liquid bubble to a solid bubble, accompanied by an expansion of the volume of actuator, as shown in Figure 4a. The McKibben type ethanol phase change actuator limits the axial expansion of the actuator due to the presence of a nylon mesh and self-locking nylon ties. The nylon mesh limits the expansion of the actuator to radial expansion and axial contraction. With the cessation of heating, the ethanol bubbles revert to liquid bubbles. Due to the presence of silicone elastomer, the ethanol phase change actuator returns to its initial state, forming a telescopic reciprocating cycle actuation, the actuation process of which is shown in Figure 4c.

To study deformation characteristics of the ethanol phase change artificial muscle, we recorded the length change in its movement process. The voltage was 8 V, and the energization time was 140 s. The deformation process is shown in Figure 4a, and the length deformation rate is shown in Figure 4b. It can be observed that the ethanol phase change actuator is gradually getting larger throughout the movement process, and the deformation can be observed obviously at 30 s. Firstly, the actuator deforms radially, at 80 s, the radial length reaches its maximum, then gradually starts to expand axially at around 80 s, and this expansion will continue as the voltage-on time increases. Combined with the length deformation rate graph, we found that the driving rate of ethanol phase change artificial muscle became faster after 30 s, and the driving process gradually smoothed out. At 100 s, the length deformation rate of the ethanol phase change artificial muscle reached a maximum value of 41.18%. The expansion rate was about 200% at an energization time of 140 s. We observed a retraction phase after 100 s. As illustrated in the deformation graph and curve, the ethanol phase change actuator is dominated by radial elongation in its initial stage, and starts to expand axially after the radial elongation reaches its limitation, mainly because the contraction of the ethanol phase change actuator is limited by the length of internal resistive wire, which makes the actuator unable to increase further after the length reaches a certain level. However, the internal ethanol continues to vaporize, so that the volume of ethanol phase change actuator increases continually, and it starts to expand along the axial direction due to the radial limitation, and this volume change is only a difference in deformation behavior, but its volume change rate is constant.

The ethanol phase change material was also made into McKibben type, and the driving process is shown in Figure 4c, and the deformation contraction curve is shown in Figure 4d. From the figure, it can be observed that the McKibben type ethanol phase change actuator is linearly contracted throughout the driving process, and the process is relatively smooth. The ethanol phase change actuator reaches the maximum deformation rate 130 s later, and the length contraction rate can reach 14.79%. After the power is cut off, the McKibben ethanol phase change actuator begins to contract gradually. At 230 s, the length recovery rate can reach 96.5%, which indicates that the McKibben ethanol phase change actuator can produce axial contraction and radial expansion, and the process is reversible. This contraction/reversal drive property is greatly similar to the motor properties of biological muscles.

It can be seen from Figure 4e that the length change of three ethanol phase change actuators is slow and then fast throughout the driving process, and the difference in deformation rate of three ethanol phase change actuators appears at 27 s. The deformation rate of the graphene oxide/ethanol phase change actuator is the largest, while the shrinkage rate of blank control and Au nanoparticle/ethanol phase change actuator is less, and the difference between them is not significant. As the energization time increases, the deformation rate gradually became larger. From the slope, it can be concluded that the addition of graphene oxide and Au nanoparticles increases the deformation rate and actuation efficiency, and the strengthen performances of graphene oxide is better than that of Au nanoparticles. It can be seen in Figure 4f, that the deformation of the ethanol phase change artificial muscle, with different graphene oxide contents, all perform axial expansion and radial contraction throughout the driving process. The deformation can be observed with the naked eye at 40 s. The deformation rate of GO-15 graphene oxide/ethanol phase change artificial muscle actuator is the largest, which can contract to half of the original length at 80 s. The deformation of whole process is slow in the initial stage and then fast. The deformation rate of ethanol phase change actuator increases after 18 s, and the degree of deformation also starts to become larger. With the increase of graphene oxide content, we can see that the deformation rate gradually increases, i.e., the slope of each line segment increases sequentially, and the driving rate of GO-15 is the fastest at 0.19 mm/s, which is 61% higher than GO-0. This indicates that the addition amount of graphene oxide affects the deformation of the graphene oxide/ethanol phase change artificial muscle actuator, causing the driving characteristics of the actuator to change. The driving rate and deformation degree of the graphene oxide/ethanol phase change artificial muscle actuator are enhanced with the increase of the graphene oxide content. The deformation analysis of the graphene oxide/ethanol phase change artificial muscle actuator verifies that the addition of graphene oxide can effectively increase its driving rate and enhance deformation capability.

### 3.3. Mechanical Properties Test

To test the mechanical properties of silicone elastomer, graphene oxide/silicone elastomer and Au nanoparticles/silicone elastomer, tensile tests were performed on various samples, as shown in Figure 5a. Each silicone elastomer sample was tested three times to obtain the stress-strain curves of three materials, as shown in Figure 5b. The maximum stresses in the tensile state for the three materials were 640 kPa, 693 kPa and 708 kPa, respectively. The fracture strain rates were 513%, 676% and 882%, respectively. The elastic moduli were 0.125 MPa, 0.103 MPa and 0.08 MPa. It is obvious that the maximum stress and fracture strain of the silicone elastomer are enhanced by the addition of Au nanoparticles and graphene oxide, but the elastic moduli decreased. With the decrease of elastic moduli, deformation will be easier, while the increase of maximum stress and fracture strain makes the material produce more deformation, which has a significant effect on enhancing the driving ability of ethanol phase change artificial muscle. Among them, graphene oxide improves the mechanical properties of the material better than Au nanoparticles.

Subsequently, we performed tensile tests on the samples with different graphene contents and the stress-strain curves were shown in Figure 5c. The maximum stresses of the silicone elastomers with different graphene oxide contents were 110 kPa, 121 kPa, 133 kPa and 150 kPa, respectively. The fracture strain rates were 615%, 634%, 665% and 621%, respectively. The moduli of elasticity were 0.224 MPa, 0.223 MPa, and 0.221 MPa. It can be observed that with the increase of graphene oxide content, the maximum stress and strain at the break of the silicone elastomer are increased, but the moduli is decreased. With the decrease of the elastic moduli, the deformation of the ethanol phase change actuator will be easier, and the increase of maximum stress and strain at the break makes the material more deformable. Therefore, with the increase of graphene oxide content, the ethanol phase change actuator is more likely to achieve large deformation.

### 3.4. Thermal Performance Analysis

As shown in Figure 6a, the heating of composites started at room temperature to above the boiling point of ethanol (≥78.32 °C). The heat absorption reaction can be observed by thermogravimetric and differential thermal analysis curves (DSC-TGA). It can be seen from the curves that the ethanol phase change composites undergo thermal decomposition when heated. The reaction rate increases when the temperature exceeds 70 °C because of the evaporation of ethanol, with a peak at 84 °C. After 84 °C, the vaporization reaction rate decreases to 0. The termination of the vaporization reaction is due to the complete consumption of ethanol in the composite.

The TGA curve can support the information reflected in the DSC curve, as it shows a slight increase in weight loss at about 55 °C, a significant weight loss from about 70 °C, and almost complete disappearance of weight loss at about 85 °C. During this process, the ethanol gradually vaporizes and escapes from the pore structure of silica gel, resulting in a reduction in the mass of the actuator by about 12% of the initial mass. Multiple iterations of the experiment showed that the actuator could be effective for 100 cycles. The DSC-TGA data showed that the most pronounced actuation would occur between 55 °C and 85 °C, with negligible action below this range. The operating temperature range of the ethanol phase change actuator is 55–85 °C (rather than room temperature to 78 °C), which can provide a very important assistance in the design of ethanol phase change actuators.

Thermal imaging test analyses were performed for blank sample, graphene oxide/ethanol phase change actuator and Au nanoparticle/ethanol phase change actuator, and the results were shown in Figure 6c. It can be seen that the heating rates of graphene oxide/ethanol phase change actuator and Au nanoparticle/ethanol phase change actuator are significantly faster than that of no adding sample. This proves that the addition of graphene oxide and Au nanoparticles can effectively accelerate the heating rates of the materials. The heating rates of the composites with graphene oxide and Au nanoparticles were the same until 52 °C. As time increases, the heating rate of composites with graphene oxide was about 10 °C higher than that with Au nanoparticles, proving that the thermal conductivity of graphene oxide was better than that of Au nanoparticles, which accelerated the deformation rate. As exhibited in Figure 6b, the heating process of three actuators is roughly linear, and the heating rates of ethanol phase change actuator, Au nanoparticle/ethanol phase change actuator and graphene oxide/ethanol phase change actuator can be calculated as 0.57 °C/s, 0.99 °C/s and 1.1 °C/s, respectively, within 100 s. The heating rates of the composites with additions are higher than that of the reference sample, and the addition of graphene oxide enhances the thermal conductivity more significantly, which is better than that of Au nanoparticles added actuator.

### 3.5. Driving Force Performance Analysis

The driving performance of graphene oxide/ethanol phase change actuator with different graphene oxide content are tested with a driving voltage of 8 V and total duration is 300 s, including a voltage-on time of 120 s and a voltage-off time of 180 s. The experimental data are shown in Figure 7a.

The output force of graphene oxide/ethanol phase change actuators started to change slowly with the increase of energization time. Due to the low temperature at the beginning, some ethanol vaporized. When energized for 38 s, the driving force of four actuators began to gradually increase, and the increase speed of the driving force became larger with the increase of graphene oxide content.

When the power was cut off at 120 s, the deformation of the ethanol phase change actuator continued, and the driving force reached its maximum potential at 130 s. From the drive force recovery curve, it can be seen that the drive force recovery of the ethanol phase change actuator is approximately a straight line during 130–180 s. After 200 s, the drive force recovery becomes slower and slower, because that the temperature of the ethanol phase change actuator starts to drop as the power is cut off. When the temperature decreases to a certain level, the cooling and liquefaction of ethanol become slower, and the slope of the force response curve becomes concave. At 280 s, the driving force of the actuators with different graphene oxide contents all become 0. This indicates that the driving force and the driving force recovery speed of the ethanol phase change actuator are increased with the increase of graphene oxide content.

Comparing the drive time and response time of the graphene oxide/ethanol phase change actuator, we found that the drive/reduction rate ratio is 1.5, which means that the ethanol phase change actuator needs to wait 1.5 times longer for its response at the end of use. This is the same as that of no-addition. The graphene oxide/ethanol phase change actuator not only has reversible deformation properties similar to the contraction/reduction of biological muscles, but also has good driving properties. Based on this property, a simple weight-lifting experiment was designed in this paper, as shown in Figure 7b. A simple weightlifting device designed by the McKibben-type graphene oxide/ethanol phase change artificial muscle actuator can lift a 50 g weight (10 times its own weight) by 12 mm in 300 s driven by 8 V, 1 A.

## 4. Conclusions

In the current work, we have successfully prepared a novel silicone/ethanol/(graphene oxide/gold nanoparticle) composite elastomer actuator for soft driving, and a composite elastomer actuator with different graphene oxide content, with their functional characteristics discussed separately.

Based on the results of microstructure, mechanical properties and mechanical driving tests, an outcome that the addition of graphene oxide and Au nanoparticles could effectively improve the actuation rate of composite elastomer actuator is demonstrated, and the enhancement effect of graphene oxide is better than that of Au nanoparticles. SEM results show that the incorporation of graphene oxide and Au nanoparticles do not affect the distribution of ethanol throughout the composite elastomer. The mechanical properties of the materials show that the addition of graphene oxide and Au nanoparticles could improve the mechanical properties of composites, reducing the hardness, and thus improving its deformation. Thermal imaging results show that the fast actuation mechanism of the actuator is to accelerate the boiling and evaporation of ethanol, which makes silicone rubber swell faster and become less hard, thus reducing the deformation time to increase actuation effect. According to our research, the best actuation effect is achieved by adding graphene oxide content of 15 mg. Our research shows that the addition of a thermally conductive enhancement phase helps to improve actuation efficiency of composite elastic actuators, which provides a direction for future development of elastic actuators to achieve multiple cycles of composite actuators. In the application of soft robots, solving the problem of internal ethanol volatilization and achieving reliable long time cycle fast actuation is one of the main directions for future research.

## Figures and Tables

**Figure 1 polymers-13-04095-f001:**
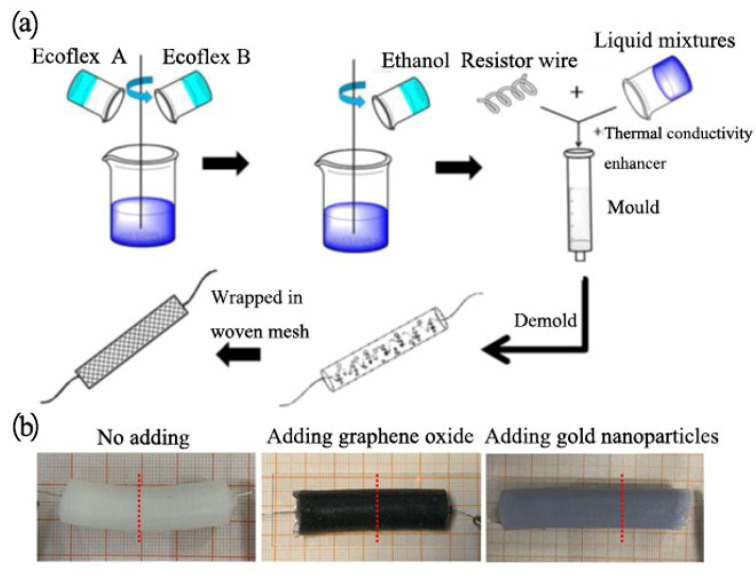
Preparation process of ethanol phase change material. (**a**) Schematic diagram of manufacturing process of phase change type actuator. (**b**) Physical actuators of different types.

**Figure 2 polymers-13-04095-f002:**
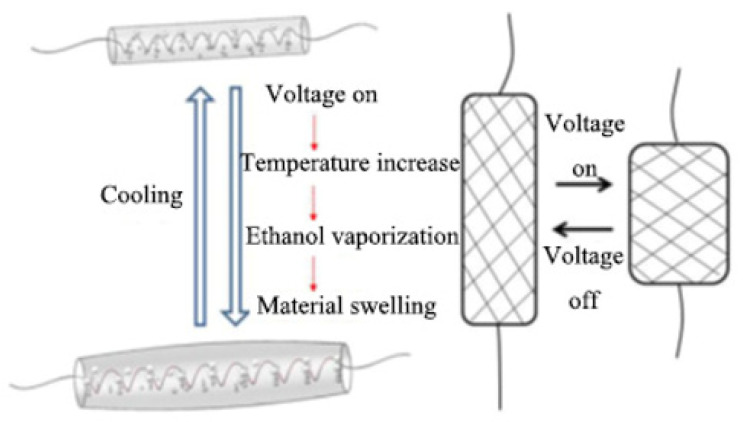
Schematic diagram of the driving mechanism of the ethanol phase change artificial muscle.

**Figure 3 polymers-13-04095-f003:**
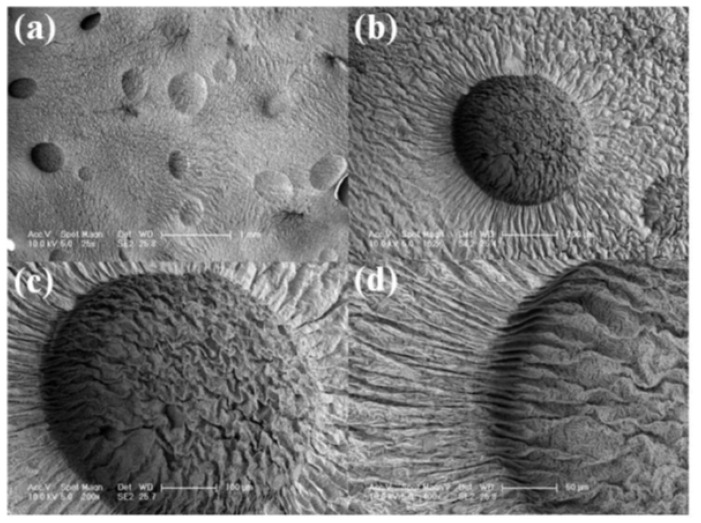
SEM image of ethanol phase change artificial muscle material under different magnifications (**a**) 25 times (**b**) 100 times (**c**) 200 times (**d**) 400 times.

**Figure 4 polymers-13-04095-f004:**
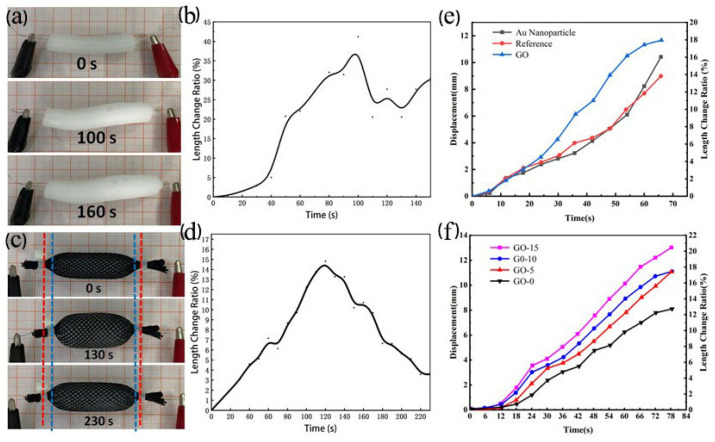
Deformation characteristics of phase-change type actuators. (**a**) Deformation process of ethanol phase change material. (**b**) Deformation rate of ethanol phase change materials. (**c**) Deformation process of McKibben-type ethanol phase change material. (**d**) Deformation rate of McKibben-type ethanol phase change material. (**e**) Deformation characteristics of different types of composite elastomeric actuators. (**f**) Deformation properties of ethanol phase change actuators with different contents of graphene oxide.

**Figure 5 polymers-13-04095-f005:**
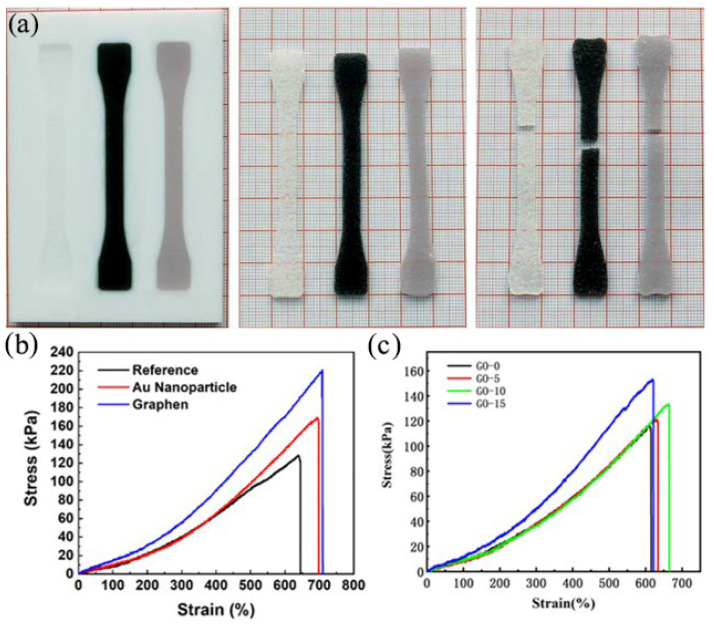
Mechanical properties of different types of ethanol phase change materials. (**a**) Tensile samples (without additions, with GO and Au nanoparticles), (**b**) Stress-strain curves of three samples. (**c**) Stress-strain curves of samples with different graphene oxide contents.

**Figure 6 polymers-13-04095-f006:**
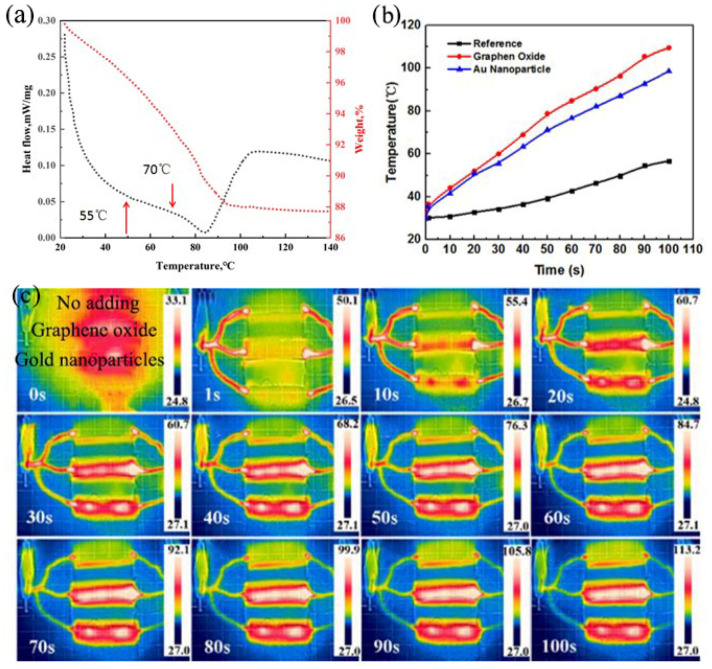
Thermal performance of the phase change type actuator. (**a**) DSC-TGA curves of material. (**b**) Temperature rises the curves of composite elastomeric actuators. (**c**) Thermographic testing of composite elastomeric actuators.

**Figure 7 polymers-13-04095-f007:**
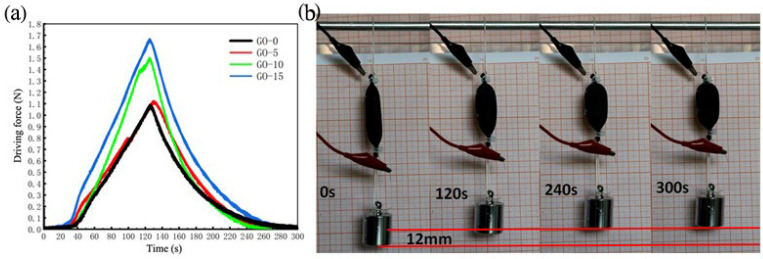
Output force characteristics of phase change type actuators. (**a**) Variable drive characteristics of curve of actuators. (**b**) Graphene oxide/ethanol phase change artificial muscle for lifting heavy objects.

**Table 1 polymers-13-04095-t001:** No addition, specific composition ratio of composite material actuator with go added and gold nanoparticles added.

Scheme	Ecoflex A (mL)	Ecoflex B (mL)	Ethanol (mL)	Graphene Oxide (mg)	Au Nanoparticle (mL)
No adding	3	3	1.2	0	0
Adding Graphene Oxide	3	3	1.2	15	0
Adding Au Nanoparticle	3	3	1.2	0	2.5

**Table 2 polymers-13-04095-t002:** Specific composition ratios of composites actuator with different contents of go added.

Sample	Ecoflex A (mL)	Ecoflex B (mL)	Ethanol (mL)	Graphene Oxide (mg)
GO-0	3	3	1.2	0
GO-5	3	3	1.2	5
GO-10	3	3	1.2	10
GO-15	3	3	1.2	15

## Data Availability

The data presented in this study are available on request from the corresponding author.

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
