# Peer review of "Ethanol Phase Change Actuator Based on Thermally Conductive Material for Fast Cycle Actuation"

_polymers, 2021, doi:10.3390/polym13234095_

Round 1
Reviewer 1 Report
The manuscript can be published in Polymers. The following comments are suggested to be considered.
- The quality of some figures needs to be improved (Figure 4b, e, d, f, Figure 6a)
- As mentioned in Table 1, a 2.5 ml solution of gold NPs has been used. What was the concentration of gold NPs in that 2.5 ml solution?
- What about the ethanol loss through the process? Please, discuss it regarding the vapor permeability of silicon-based elastomers.
Author Response
Dear reviewer
Thank you for your comments on this article and for your valuable comments.In response to your questions, we have made corrections and additions, and we will answer them one by one below.
1.For the low-resolution images in the manuscript, we have replaced them with high-resolution images.
2.Regarding the gold nanoparticle concentration, it has a concentration of 1mol/L, which we have added in the paper.
3.Regarding the loss of ethanol, thermogravimetric and differential thermal analysis tests (DSC-TGA) of the composites were performed. The composites were heated from room temperature to above the boiling point of ethanol (≥78.32°C). The heat absorption reaction can be observed by DSC-TGA curves. It is evident from the curves that the ethanol phase change composites undergo thermal decomposition upon heating. When the temperature exceeds 70°C, the reaction rate increases due to the evaporation of ethanol, with a peak at 84°C. After 84°C, the vaporization reaction rate decreases to 0. The termination of the vaporization reaction is due to the complete consumption of ethanol in the composites. The weight loss of the material increases slightly at about 55°C, the weight loss is obvious from about 70°C, and the weight loss almost completely disappears at about 85°C. During this process, the ethanol gradually vaporizes and escapes from the pore structure of the silica gel, resulting in a reduction of the mass of the actuator by about 12% of the initial mass. Multiple iterations of the experiment showed that the actuator could be effective for 100 cycles. the DSC-TGA data showed that the most pronounced actuation would occur between 55°C and 85°C, with negligible effects below this range. The operating temperature range of the ethanol phase change actuator is 55°C-85°C (rather than room temperature to 78°C), which can provide very important assistance in the design of ethanol phase change actuators.

Reviewer 2 Report
The manuscript novalety is unclear; for example what is the difference between the currecnt research and the previously published “Xia, Boxi, Aslan Miriyev, Cesar Trujillo, Neil Chen, Mark Cartolano, Shivaniprashant Vartak, and Hod Lipson. "Improving the actuation speed and multi-cyclic actuation characteristics of silicone/ethanol soft actuators." In Actuators, vol. 9, no. 3, p. 62. Multidisciplinary Digital Publishing Institute, 2020.”
Title must be revised to reflect the significance of your research. Current title is very primitive.
Similarly, abstract is written very badly and must be revised.
Authors should discuss in introduction section the reason for choosing
silicone/ethanol/(graphene oxide/gold nanoparti-cle).
References not enough and must be updated.
The action mechanism of ethanol phase change artificial muscle .
Language must be revised, streams of statements need technical revisions.
Author Response
Dear reviewer
Thank you for your valuable suggestions on this article.In response to your questions, we have made changes and additions in the article and explained them in detail below.
1.In response to your comment about the article entitled “Improving the actuation speed and multi-cyclic actuation characteristics of silicone/ethanol soft actuators”,we have read it. The difference between this paper and the above article is that we have added different thermally conductive materials to the ethanol phase change material and demonstrated that graphene oxide can increase the deformation rate of the actuator faster.When the voltage is 8V, the temperature rise rate can reach 1.1℃/s.
2.The title of the article has been changed to “Ethanol phase change actuator based on thermally conductive material for fast cycle actuation“.
3.Likewise, the abstract has been modified in the paper.
4.Ecoflex silicone elastomer is a kind of silicone rubber material commonly used for making soft robots, which has good flexibility and mechanical properties, and the model used herein is Ecoflex00-50. As a very common organic compound in daily life, ethanol is a flammable, volatile, colorless and transparent liquid at room temperature and pressure with a boiling point of 78°C. It is easy to realize vaporization after giving external heat and liquefy to its initial state when the temperature decreases. The main source of deformation of the ethanol phase change drive is gas/liquid change of ethanol, which is responsible for the volume change of the elastomer composites.Owing to the good electrothermal conversion rate, graphene oxide and Au nanoparticles were adopted as thermally conductive enhanced ingredients, respectively.
5.The reference has been updated.
6.The ethanol in the ethanol phase change actuator is stored and dispersed as a liquid bubble in the silicone elastomer. When the power is turned on, the resistive wire in the ethanol phase change actuator starts to be heated. As the temperature rises, the ethanol bubble begins to gradually change from a liquid bubble to a bubble, accompanied by an expansion of the volume of actuator, as shown in Figure 4a. The McKibben type ethanol phase change actuator limits the axial expansion of the actuator due to the presence of a nylon mesh and self-locking nylon ties. The nylon mesh limits the expansion of the actu-ator to radial expansion and axial contraction. With the cessation of heating, the ethanol bubbles revert to liquid bubbles. Due to the presence of silicone elastomer, the ethanol phase change actuator returns to its initial state, forming a telescopic reciprocating cycle actuation, the actuation process of which is shown in Figure 4c.
7.The phrases in the article have been rewritten.
